# Molecular Advances in Sinusoidal Obstruction Syndrome/Veno-Occlusive Disease

**DOI:** 10.3390/ijms24065620

**Published:** 2023-03-15

**Authors:** Ioulia Mavrikou, Dimitrios Chatzidimitriou, Lemonia Skoura, Emmanouil Nikolousis, Ioanna Sakellari, Eleni Gavriilaki

**Affiliations:** 1Hematology & BMT Unit, General Hospital “George Papanikolaou”, 57010 Thessaloniki, Greece; 2Microbiology Department, Aristotle University of Thessaloniki, 54124 Thessaloniki, Greece; 3Department of Microbiology, AHEPA University Hospital, Medical School, Faculty of Health Sciences, Aristotle University of Thessaloniki, S. Kyriakidi Str., 54636 Thessaloniki, Greece; 4Department of Haematology, University Hospitals Birmingham, Birmingham B9 5SS, UK; 5School of Medicine, Aristotle University of Thessaloniki, 54124 Thessaloniki, Greece

**Keywords:** SOS/VOD, HSCT, defibrotide, complement activation, endothelial injury syndromes

## Abstract

Sinusoidal obstruction syndrome/veno-occlusive disease (SOS/VOD) detected in the liver has been considered a severe complication of hematopoietic stem cell transplantation (HSCT). SOS/VOD is characterized by hepatomegaly, right upper quadrant pain, jaundice, and ascites. The severe forms of the disease may result in multi-organ dysfunction (MOD) with a high mortality rate (>80%). The development of SOS/VOD can be rapid and unpredictable. Therefore, early identification and severity assessment is crucial in facilitating prompt diagnosis and timely treatment. Effective treatment and potential prophylaxis with defibrotide highlight the need for characterizing a sub-group of patients at high risk for SOS/VOD. Moreover, antibodies that are conjugated with calicheamicin, gemtuzumab, and inotuzumab ozogamicin, have led to renewed interest in this syndrome. Evaluation and management of serious adverse events associated with gemtuzumab and inotuzumab ozogamicin are recommended. We review hepatic-, transplant- and patient-related risk factors, criteria for diagnosis and grading classification, and SOS/VOD potential biomarkers. Furthermore, we examine pathogenesis, clinical presentation, diagnostic criteria, risk factors, prophylaxis, and treatment of SOS/VOD occurring post HSCT. Moreover, we aim to provide an up-to-date summary of molecular advances in the diagnosis and management of SOS/VOD. We performed a comprehensive review of the literature and examined the recently available data, mostly using the PubMed and Medline search engines for original articles published over the last decade. In the era of precision medicine, our review provides up-to-date knowledge of genetic or sera markers for SOS/VOD with the goal of identifying a subset of high-risk patients.

## 1. Introduction

Traditionally, SOS/VOD disease of the liver has been considered a severe complication of allogeneic Hematopoietic Cell Transplantation (HCT), mainly in patients with known risk factors [1]. Jacobs et al. first described hepatic SOS/VOD and its association with HSCT [2]. Clinically, SOS/VOD is presented as the consequence of endothelial cell injury which leads to loss of sinusoidal wall integrity, endothelial cell detachment, sinusoidal obstruction, and development of postsinusoidal portal hypertension [3]. Generally, it occurs within 21 days after transplant, while late-onset SOS/VOD is currently acknowledged as a distinct SOS/VOD feature by recent diagnostic criteria elaborated by the European Society for Blood and Marrow Transplantation (EBMT) [4]. Late SOS/VOD occurs in at least 39.3% of adult HCT recipients [5]. The mortality rate of severe SOS/VOD where multi-organ dysfunction (MOD) occurs is high, reaching a percentage of more than 80% [4,6].

Several different factors provoke sinusoidal endothelial cell (SEC) damage that results in hepatic sinusoids and central vein obstruction. Clinical features of SOS/VOD include rapid weight gain, tendentially unresponsive to diuretics, hyperbilirubinemia, painful hepatomegaly, and ascites. According to the onset of SOS/VOD, it can be either smoldering or disruptive, ranging from mild forms that spontaneously resolve within a few weeks to severe forms which trigger organ damage and multiorgan failure. Multiorgan disease, in general, involves pulmonary and renal functions, with its occurrence rapidly worsening the outcome [4].

Renewed interest in this syndrome has led calicheamicin-conjugated antibodies, gemtuzumab, and inotuzumab ozogamicin. They have been approved for the treatment of newly diagnosed CD33 + acute myeloid leukemia (AML) and B-cell acute lymphoblastic leukemia (B-ALL), respectively, yet they are associated with increased risk of hepatotoxicity and SOS/VOD.

Effective treatment and, potentially, prophylaxis with defibrotide highlights the need to characterize a sub-group of patients at high risk for SOS/VOD. Advances in other endothelial injury syndromes may facilitate better understanding of SOS/VOD [7,8,9,10,11]. Additionally, research should focus on the gut microbiome and investigate more the association between rich gut microbiome before HSCT and reduced possibility of SOS/VOD development [12]. 

Therefore, we aim to provide an up-to-date summary of knowledge on advances in SOS/VOD. To this end, a comprehensive review of the literature was performed, which critically examined all recent available data, mostly using the PubMed and Medline search engines for original articles that had been published over the last decade. We provide current diagnosis criteria, risk stratification, management, prevention, prophylaxis, and treatment of SOS/VOD after HSCT.

## 2. Pathophysiology

Chemotherapy or radiotherapy may injure vascular-derived endothelial cells (ECs) and activate them. Activated ECs release heparinase, which breaks down extracellular matrix proteins. The cytoskeletal architecture is disrupted, losing its inter-cellular tight junctions, and ECs become rounded, creating intercellular gaps. Vascular permeability is increased, allowing fluid and other blood elements to leak into the extravascular space of Disse and surrounding tissues, promoting sinusoidal narrowing. Vascular endothelial activation triggers the release of multiple factors that regulate coagulation and fibrinolysis (Figure 1).

In addition, procoagulant proteins von Willebrand Factor (vWF) and plasminogen activator inhibitor-I (PAI-I) are released from the damaged endothelial cells. Thrombocyte aggregation and clot formation are promoted by vWF whereas fibrin lysis is inhibited by PAI-I. These coagulant states result in microvascular clot formation in hepatic sinusoids and venules and post-sinusoidal obstruction [13,14]. Accordingly, blood flow is obstructed, and hepatic outflow is reduced, producing post-sinusoidal portal hypertension (PH) [15]. Post-sinusoidal PH triggers clinical signs and symptoms of VOD/SOS.

## 3. Clinical Presentation, Diagnostic Criteria and Risk Stratification

The clinical manifestations of SOS/VOD are characterized by weight gain (unresponsive to diuretics), hyperbilirubinemia, painful hepatomegaly, and ascites. Traditionally, SOS/VOD is diagnosed according to the Baltimore or modified Seattle criteria (Table 1), after the exclusion of other conditions [16,17]. Numerous conditions, such as constrictive pericarditis, fluid overload, ascites of different origin (pancreatic, chylous), drug-induced cholestasis and, more generally, drug-induced liver injury (DILI), sepsis, cholestasis, infectious hepatitis, cholangitis lenta, parenteral nutrition and hepatic graft-versus-host disease (GVHD), imitate SOS/VOD, making real-life differential diagnosis a true challenge. Between the two traditional criteria, the main difference is hyperbilirubinemia, which is mandatory in the Baltimore criteria and implies a longer time waiting for its development or intrinsically more aggressive forms. Clinically the scenario varies, and changes dynamically, especially in the pediatric population [18]. Therefore, new different diagnostic criteria and a scale for the severity grading of suspected SOS/VOD are proposed by EBMT. For adult patients, the EBMT criteria foresee two clinical entities: the classical SOS/VOD, appearing within 21 days after HSCT with bilirubin ≥2 mg/mL, and two of the following criteria: painful hepatomegaly, weight gain, and ascites (Table 1). There is an association between the EBMT criteria and severity grading scales that are related to the dynamic changes, mainly the evolution of hepatic and renal function tests. A warning sign belonging to a higher severity grading scale (for suspected SOS/VOD) is considered the speed of changes, therefore supporting early treatment initiation with potential clinical outcome improvement. Concerning grading SOS/VOD severity in adult patients, new EBMT criteria are proposed. A grading scale is established where the kinetics of the appearance of symptoms, the sooner bilirubin, transaminases, and weight increase, the more severe the SOS/VOD is characterized [4]. In case of suspected SOS/VOD before day 21 and particularly before patients meet the diagnostic criteria, this scoring system can be used [4]. An early diagnostic criterion for hepatic VOD in children is Refractory Thrombocytopenia (RT), which is included in the 2017 EBMT criteria [18,19].

Clinical criteria are essential in order to establish an SOS/VOD diagnosis; nevertheless, there arise many interpretation problems. The gold standard method for diagnosis and differential diagnosis of SOS/VOD is liver biopsy, although this is an invasive technique that carries a high risk of bleeding and infection complications [1,20].

One of the most accurate imaging techniques for SOS/VOD diagnosis is hepatic vein pressure gradient (HVPG) measurement, which predicts and diagnoses via the assessment of the degree of PH. However, using it in routine practice cannot be considered, since it is an invasive technique [4,21].

The diagnosis of SOS/VOD using ultrasound (US) is controversial due to the nonuniformity of the results; therefore, US imaging may be helpful for the elimination of other etiologies that mimic SOS/VOD [20,21].

Computed tomography has been studied for the diagnosis of SOS/VOD and has been shown to be beneficial for differentiating SOS/VOD from GVHD [22].

The diagnosis of SOS/VOD is followed by an evaluation of disease severity to identify which patients require early therapeutic intervention. EBMT recently published a severity scoring system for SOS/VOD in adult patients, shown in Table 2 [4].

The development of risk stratification for SOS/VOD provides the opportunity for the early diagnosis and management of the disease. Additionally, acknowledging the importance of identifying risk factors in order to facilitate prompt diagnosis and timely treatment, recent studies have emphasized patient-related factors. The Center for International Blood and Marrow Transplant Research (CIBMTR) supports a method of SOS/VOD risk calculation/scoring that widely recognizes risk factors as age, Karnofsky Performance Status score, sirolimus use, hepatitis B/C status, type of conditioning regimen, and primary disease associated with HSCT [23]. These independent prognostic factors for the development of SOS/VOD by day +100 after HCT are identified using a multivariate logistic regression model. The calculator has been tested in 13,097 HSCT recipients in the CIBMTR database and has been shown to stratify risk levels among patients. According to their risk score percentile, patients are stratified into four distinct, statistically significantly different groups. This pretransplant risk score successfully stratifies allogeneic HCT patients by risk of developing SOS/VOD. Independent prognostic factors include younger age, use of sirolimus, lower Karnofsky performance scale score, positive hepatitis B/C serology, disease, disease status at transplant, and conditioning regimen. By day +100, 637 patients (4.9%) developed VOD. Higher risk indicated myeloablative conditioning regimens and busulfan-based myeloablative conditioning regimens guided by pharmacokinetic monitoring.

Moreover, a standard biomarker panel has been developed to assess endothelial dysfunction and activation, cited Endothelial Activation and Stress Index (EASIX), based on the formula “lactate dehydrogenase (LDH) (U/L) * creatinine (mg/dL)/thrombocytes (10^9^ cells per L)” [24]. Jiang et al. (2021) compared the potential of EASIX with that of the SOS/VOD CIBMTR clinical risk score to predict SOS/VOD in two independent cohorts. EASIX showed a statistically significant association with SOS/VOD incidence and has been proven to be a valid biomarker for defining a subpopulation of allogeneic stem cell transplantation (allo-SCT) recipients at high risk for SOS/VOD.

Evidence suggests the association of gut microbiome (GM) with HSCT outcomes [25]. It is well known that GM plays a vital role in the pathophysiology of GvHD and bloodstream infection (BSI) in the pediatric population [26,27]. Pre-transplant GM signatures such as low diversity, low number of beneficial microbes, and increased proportion of pathogens and pathobionts, regulate metabolic and immunological homeostasis, and may predict outcomes and HSCT complications [28]. Therefore, a retrospective case–control study was conducted in allo-HSCT pediatric patients, investigating the development or not of SOS/VOD, while simultaneously examining their GM profile over time, from before the transplant up until 72 days after [12]. The findings showed significantly lower alpha diversity and depletion in typically health-associated taxa compared to controls, both in pre-transplant and post-transplant samples. In addition, patients who did not develop SOS/VOD presented enhanced health-associated commensals such as Bacteroides, Ruminococcaceae, and Lachnospiraceae. Consequently, this study states the hypothesis that a rich and diverse GM before HSCT appears to be associated with a reduced possibility of developing SOS/VOD.

Ozkan et al. (2022) intended with a retrospective study, to examine the role of liver stiffness measurement (LSM) with transient elastography (TE) for the diagnosis of SOS/VOD after allogeneic HSCT [29]. A total of 31 adult patients who had two or more LSM values participated in the study. Of the 31 participants, two patients (6.4%) developed SOS/VOD. Their clinical evaluation indicated very high LSM values; thus, TE could be a promising non-invasive imaging method for SOS/VOD diagnosis. Due to the small sample, further research needs to be performed.

## 4. Risk Factors

Recognition of SOS/VOD risk factors facilitates expedited treatment. The latest studies analyzing the risk factors for SOS/VOD in large HSCT recipient populations vary in consistency.

Corbacioglu et al. (2019) investigated 10 studies with total numbers of patients ranging from 75 to 5072 with an incidence of SOS/VOD ranging from 2.0% to 30.7% [30]. According to this review, transplantation-related factors associated with increased risk of SOS/VOD in adult HSCT recipients include oral administration of busulfan, use of horse anti-thymocyte globulin, previous radiation therapy, increased trough serum tacrolimus levels, myeloablative conditioning (MAC) regimens, and two or more HSCTs. In children, an independent risk factor for SOS/VOD is the use of busulfan in conditioning regimens. 

Moreover, data were collected from all adult patients who underwent MAC HSCT at the Dana-Farber/Brigham and Women’s Cancer Center between 1996 and 2015 (*n* = 205) to identify clinical parameters that could lead to early detection of SOS/VOD [31]. In the period from 7 days before and the day of diagnosis, SOS/VOD patients were observed to have high serum creatinine levels and were likely to develop acute kidney injury, experienced refractoriness to platelet transfusion, and had high trough serum tacrolimus levels. A study concerning patient/hepatic-related SOS/VOD risk factors indicated Eastern Cooperative Oncology Group Performance Status score 2 to 4 versus 0 to 1, hepatitis C seropositivity, and advanced disease status [32], while another one included high pre-HSCT ferritin level (≥950 ng/mL) in patients with malignant lymphoma [33].

Furthermore, gemtuzumab ozogamicin (GO) is an antibody–drug conjugate containing a CD33-directed monoclonal antibody that is covalently linked to the cytotoxic agent N-acetyl gamma calicheamicin [34]. The CD33 antigen is expressed on most myeloid leukemic blasts and immature normal cells of myelomonocytic lineage, but not on normal hematopoietic stem cells. Gemtuzumab is a monoclonal antibody, which has been designed to attach to CD33 on leukemia cells. Cells absorb the antibody and the cytotoxic substance calicheamicin, which eventually breaks up the cells’ DNA.

In 2017, GO was approved by the US Food and Drug Administration (FDA) for the treatment of newly diagnosed CD33 + AML in adults in combination with chemotherapy, or as a single agent in relapsed or refractory (R/R) CD33 + AML in adults and pediatric patients ≥2 years old [35]. In 2020, GO was included in combination with standard chemotherapy for newly diagnosed pediatric patients ≥1-month-old [36].

Following HSCT, GO treatment has been associated with an increased risk of hepatotoxicity and SOS/VOD. The pathophysiology of GO-associated SOS/VOD may result from the delivery of calicheamicin to CD33 + sinusoidal endothelial cells, as such cells are known to express this surface protein [37].

A randomized, open-label phase III ALFA-0701 study, undertaken among 26 hematology centers, in patients with previously untreated de novo acute myeloid leukemia, analyzed how efficient and safe the administration of smaller, divided doses of GO could be in combination with standard chemotherapy [38]. Findings demonstrated that the use of fractionated, lower doses of GO granted safe delivery of higher cumulative doses and improved outcomes in patients with acute myeloid leukemia [39].

Similarly, inotuzumab ozogamicin (IO), an anti-CD22 calicheamicin-linked monoclonal antibody that is approved for B-ALL, has been identified as a crucial risk factor for drug-induced liver injury and SOS/VOD [40]. Inotuzumab is a monoclonal antibody that has been designed to attach to CD22, a protein found on the surface of B cells. Cancerous B cells absorb the antibody, and calicheamicin becomes active and breaks up the cells’ DNA.

Before the presence of classical diagnostic criteria, these dynamic clinical markers could alert clinicians to the development of SOS/VOD.

## 5. Prophylaxis

Given the low survival rates of SOS/VOD patients, there is an unmet clinical need for pharmacological prophylaxis. Several approaches have been studied, including heparin, prostaglandin E1, antithrombin, pentoxifylline, and ursodeoxycholic acid [41]. Our group has studied the effect of heparin and fresh frozen plasma in a randomized study, failing to show benefits compared to fresh frozen plasma alone [41]. Furthermore, a meta-analysis of 12 studies that used unfractionated heparin (UFH) or low-molecular-weight heparin (LMWH) as prophylaxis for SOS/VOD did not significantly reduce the risk [42].

### 5.1. Prostaglandin E1

There is a small number of reports studying the use of prostaglandin E1 in the prevention of SOS/VOD. Nevertheless, severe side effects and toxicities have been reported by groups of patients where prostaglandin E1 was used [43,44].

### 5.2. Pentoxifylline, Antithrombin

Pentoxifylline has been used in the prophylaxis of regimen-related toxicities that follow stem cell transplant, but this did not prove beneficial for SOS/VOD treatment [45]. Neither did antithrombin [46]. 

### 5.3. Ursodeoxycholic Acid (UDCA)

Among prophylactic measures, only ursodeoxycholic acid is currently recommended by the guidelines of the British Society of Haematology [20]. Ursodeoxycholic acid (UDCA) is a secondary bile acid with cytoprotectant, immunomodulating and choleretic effects. It is used in the management and treatment of cholestatic liver disease, reducing the cholesterol fraction of biliary lipids. Tay et al. performed a systematic review of six studies, confirming that the use of prophylactic UDCA significantly reduced SOS/VOD in patients who underwent allogeneic HSCT [47]. According to the British Committee for Standards in Haematology (BCSH) and the British Society for Blood and Marrow Transplantation (BSBMT), UDCA is recommended for the prevention of SOS/VOD [20], while patients who received UDCA indicated a lower risk of veno-occlusive disease (VOD) [19].

### 5.4. Defibrotide

A really interesting possibility would be the use of defibrotide as a prophylactic agent. Defibrotide is a complex single-stranded oligodeoxyribonucleotids with fibrinolytic properties, which is approved for the treatment of both adult and pediatric patients with SOS/VOD with renal or pulmonary dysfunction following HSCT, and has demonstrated protective effects on micro- and macrovascular endothelium [48]. Except for several retrospective studies, only one phase 3, randomized, open-label, multicenter trial has been conducted in the pediatric setting [49]. Enrolled patients had one or more SOS/VOD risk factors, including preexisting hepatic disease, second myeloablative transplant, allogeneic transplant for leukemia beyond second relapse, conditioning with busulfan and melphalan, prior treatment with gemtuzumab ozogamicin, or a diagnosis of primary hemophagocytic lymph histiocytosis, adrenoleukodystrophy, or osteopetrosis. In the defibrotide group, twenty-two patients (12%) developed SOS/VOD in comparison with 35 patients (20%) in the control group (*p* = 0.048). The British Committee for Standards in Haematology and the British Society for Blood and Marrow Transplantation guidelines based on these results recommend the use of defibrotide for SOS/VOD prophylaxis in children undergoing HSCT with at least one risk factor for SOS/VOD [28]. In adults, there is a completed phase 3, randomized trial of 372 participants, where the efficacy and safety of defibrotide were studied contrary to the best supportive care concerning VOD prevention, and VOD-free survival measured by day +30 post-HSCT in patients who were at high or very high risk (ClinicalTrials.gov NCT02851407). In the defibrotide group, 66.8% of 190 participants were VOD-free by day +30 post-HSCT compared to 72.5% of 182 participants in the best supportive care group.

## 6. SOS/VOD Management

### 6.1. Defibrotide

Additionally, specific therapy with defibrotide management includes supportive and intensive care. Defibrotide is the only approved drug for the treatment of moderate/severe SOS/VOD. Although its mechanism of action has not yet been fully clarified, it exerts fibrogenetic as well as angiogenetic effects with endothelial stabilization [50,51]. Furthermore, defibrotide acts as an antithrombotic and profibrinolytic drug, reducing platelet adhesion and activation without systemic anticoagulant effects by inhibiting PAI-1, thrombin, P-selectin expression, and leukocyte adhesion. It also decreases vascular permeability and apoptosis due to calcineurin inhibitors and chemotherapy, without interfering with the antitumor effect of cytotoxic drugs [52]. The efficacy and safety of defibrotide in the setting of SOS/VOD have been well established. The first study was a tedious, historically controlled multicenter open-label phase 3 study. The study demonstrated both 100-day survival and complete remission (CR) benefit, favoring the defibrotide arm. Adverse events primarily involved hemorrhagic events, but were similar in both arms [53]. In addition, 1169 patients were enrolled in the international compassionate use program. Data were finally retrieved from 710 patients [54]. These results led to FDA approval of defibrotide for the treatment of adult and pediatric SOS/VOD in 2016.

More recently, a prospective open-label, single-arm study in an expanded access program highlighted a significant relation between earlier initiation of defibrotide treatment and higher day +100 survival. Treatment-emergent adverse events also primarily involved hemorrhagic events [5]. An ongoing phase 4 study also provided interim real-world data in 2019. Interestingly, although there was no registration of defibrotide for this indication, patients who were treated with defibrotide as a prophylaxis were included. In summary, the recent systematic review of studies showed a 100-day survival of 41% in patients with multiorgan disease and 71% in those without multiorgan disease [55].

A study conducted by Strause et al. published data from 8342 patients from the CIBMTR database from 2008 until 2011, in which patients were divided into two groups, those treated with defibrotide (*n* = 41) and those not treated with defibrotide (*n* = 55) in a 3.2% incidence of SOS/VOD and a 1.2% incidence of SOS/VOD with MOD [56]. Day +100 OS was 39% in patients receiving defibrotide compared with 30.9% in those who were not. 

### 6.2. Tissue Plasminogen Activator (TPA)

Some studies have used thrombolytic therapies as a tissue-plasminogen activator (t-PA), but have failed to show efficiency [57,58]. A retrospective study reviewed 56 patients who received variable initial dose t-PA [59]. Patients were divided into two groups according to the maximum daily dose of t-PA. The response rate was higher in lower t-PAmax doses both in moderate and severe SOS/VOD, while the higher the t-PAmax dose was, the higher the mortality and bleeding complications.

### 6.3. n-Acetyl-l-cysteine (NAC)

The use of n-acetyl-l-cysteine (NAC) for early liver toxicity after allogeneic HSCT in adults did not manage to show a difference between the group that used NAC and the group that did not use it [60]. Nevertheless, in pediatric patients, complete response was achieved after diagnosis with SOS/VOD and treatment with NAC [61].

Due to the lack of efficiency and the associated risk of hemorrhage, neither tissue plasminogen activator nor NAC is recommended for the treatment of SOS/VOD by the BCSH/BSBMT guidelines [20].

### 6.4. Recombinant Human Soluble Thrombomodulin Alpha (rhTM)

A retrospective study evaluated the therapeutic potential of recombinant human soluble thrombomodulin alpha (rhTM) for SOS/VOD in 39 patients [62]. According to the study, 28 patients (72%) were diagnosed as late-onset SOS/VOD at the median day of 44, 33 patients (85%) developed severe SOS/VOD, while the median duration of rhTM management was 11 days. One year after the administration of rhTM, the incidence of complete response resolution of all symptoms and signs of SOS/VOD diagnostic criteria was 33.3%, an element that poses rhTM as possessing possible therapeutic potential for SOS/VOD.

## 7. Complement Activation in SOS/VOD

Clinical features of SOS/VOD have similarities with Hemolysis, Elevated Liver enzymes, and Low Platelet number syndrome (HELLP), a syndrome observed during pregnancy. Functional and genetic evidence has been provided by our team and other groups that points towards increased complement activation, which is associated with complement-related germline mutations in patients with HELLP syndrome [7,8,9,10,11]. Preliminary evidence of complement activation in patients with SOS/VOD has been suggested by earlier studies. Increased complement activation markers at levels similar to those of patients with transplant-associated thrombotic microangiopathy (TA-TMA) have been shown in a subset of transplanted patients with SOS/VOD. Additionally, in patients with SOS/VOD, A Disintegrin and Metalloproteinase with Thrombospondin motifs (ADAMTS13), a known regulator of TMAs was reported to be lower [63]. In line with these data, a former case report documented increased complement activation in a SOS/VOD patient who was efficiently treated with the complement inhibitor C1 esterase inhibitor (C1-INH-C) [64]. Concerning genetic studies, two complement factor H (CFH) variants were detected in three SOS/VOD patients by Bucalossi et al. No other complement-related genes have been studied except for complement factor I (CFI) [65]. Given the scarcity of available data, we aim to prospectively study the genetic susceptibility of complement dysregulation in patients with SOS/VOD and associate data from genetic and functional complement assays with clinical characteristics and outcomes. These results are expected to revolutionize the genomic landscape of the disease, identifying a subset of high-risk patients that would benefit from prophylactic VOD treatment in the setting of HCT or during treatment with novel agents.

### Advances in Other Endothelial Injury Syndromes

Other than SOS/VOD, various endothelial injury syndromes, including TA-TMA or HSCT-TMA, and GVHD, can result following allogeneic HCT. Recent advances in their understanding may facilitate progress in treatment of SOS/VOD [3].

Another life-threatening complication of HCT is TA-TMA, which manifests with microangiopathic hemolytic anemia, thrombocytopenia, and organ damage, including renal or neurologic manifestations [66,67,68,69,70,71]. It is mostly observed in post-allogeneic HCT, but has also been shown after autologous HCT, primarily in pediatric recipients [72]. Although the diagnosis is based only on routine laboratory values, the high incidence of cytopenias and organ dysfunction following allogeneic HCT complicate TA-TMA diagnosis. Renal and neurologic manifestations may present due to several other causes post-HCT [73,74,75]. Endothelial injury is a common denominator in many of these causes, including in TA-TMA. Other underlying causes (conditioning regimen, calcineurin inhibitors, infections, and GVHD) contribute to a prothrombotic state, which may finally lead to microvasculature thrombosis [76]. TA-TMA diagnosis is constantly improving based on a novel consensus prompted by a better understanding of the pathophysiology of this syndrome [77].

Recent advances, mainly in TMAs, have highlighted the role of complements in endothelial injury syndromes [78]. More than that, the list of complement-mediated diseases or complementopathies is constantly growing, as we have recently described [79]. Jodele et al. first showed that TA-TMA is characterized by endothelial dysfunction after multiple triggers in pediatric patients with genetic predisposition [71,80]. Early studies have reported excessive activation of the terminal complement pathway measuring soluble C5b-9 as a rough marker of terminal complement activation [71]. Further data have confirmed complement activation on the cell surface through functional assays [81]. Additional genetic data have suggested the occurrence of susceptibility due to rare complement-related mutations or variants, previously described in complementopathies [80]. Our group confirmed both functional and genetic complement dysregulation in adults [82], additionally suggesting a vicious cycle of endothelial dysfunction, hypercoagulability, neutrophil, and complement activation [83]. A more recent transcriptome analysis in pediatric patients has shown a complement–interferon interplay that perpetuates endothelial injury [84]. These data support phenotypic similarities of complement-mediated TMA with hemophagocytic lymphohistiocytosis (HLH), characterized by signs and symptoms of extreme inflammation [85]. Our collaborative group has studied patients with TA-TMA and supported previous evidence of complement activation, efficiently treated with complement inhibition in this severe HCT complication. In this context, these syndromes resemble the prototype model of complement-mediated TMA or hemolytic uremic syndrome (HUS) [78]. The latter is a TMA with excessive complement activation characterized by germline variants in complement-related genes [86]. Different phenotypes may be caused by different variants in these factors, as suggested in C3G-glomerulopathy and age-related macular degeneration [87,88]. Our understanding of TA-TMA pathophysiology has led to a revolution in therapeutics. Prompted by efficacy and safety in complement-mediated TMA, complement inhibitors have also shown success in TA-TMA [89,90]. The first-in-class terminal complement inhibitor eculizumab has long been used in TA-TMA [91,92,93,94]. Real-world data suggest that early initiation of treatment in patients with complement activation measured by soluble C5b-9 levels, as well as monitoring of treatment and dose adjustments, yield better results [95]. Importantly, narsoplimab, a lectin pathway inhibitor targeting Mannan-binding lectin-associated serine protease-2 (MASP-2), has received breakthrough FDA designation, based on positive data in TA-TMA [96]. As compassionate use of narsoplimab continues, additional complement inhibitors that are indicated for other complement-mediated diseases are under study: ravulizumab and pegcetacoplan.

Since defibrotide protects endothelium from toxic, inflammatory and ischemic damage, its potential therapeutic use has been shown in several endothelial disorders, such as thrombophlebitis, post-surgery deep vein thrombosis prophylaxis, and peripheral arterial diseases. It has also been administrated in acute myocardial infarction, in post-thrombolysis re-occlusion of coronary arteries, ischemic damage of the liver, diabetic microangiopathy, and Reynaud phenomenon [97]. Interestingly, defibrotide has been studied in both adult and pediatric TA-TMA [98,99]. Both studies have shown encouraging results. Given the unmet need for combination treatment in TA-TMA, defibrotide might be further considered in combination with complement inhibitors [95].

Among allogeneic HCT survivors without relapse or secondary malignancy, GVHD is the major cause of morbidity and mortality [100,101]. Treatment of GVHD consists mostly of immunosuppressive agents [102]. Prolonged immunosuppression is a risk factor for severe infections and leads to a vicious cycle of morbidity in GVHD patients [74,103,104]. Endothelial dysfunction markers such as endothelial microvesicles are significantly increased 2–3 weeks post allogeneic HCT, as well as in patients with acute GVHD [105,106,107]. Endothelial activation has also been implicated in the pathophysiology of acute GVHD by a recent experimental study [108].

## 8. Conclusions

Since there are, as yet, no validated biomarkers for SOS/VOD, diagnosis is based on clinical criteria. Nevertheless, many other conditions are able to meet the SOS/VOD criteria. Identification of at-risk patients is challenging, leading to delays in diagnosis and poor outcomes. Additional studies enrolling a higher number of patients are necessary to determine the role of immunosuppression in the pathogenesis of SOS/VOD. Further investigations are proposed regarding novel therapies using low-molecular-weight heparin, antithrombin III, tissue plasminogen activator (TPA), n-acetyl-l-cysteine (NAC), and antithrombotics [109].

Patients receiving GO or IO should be monitored closely for the development of SOS/VOD. Defibrotide is the only drug approved for the treatment of SOS/VOD. Ursodeoxycholic acid is recommended for prophylaxis.

Our up-to-date review summarized molecular advances in SOS/VOD. A case report verified the association between complement activation and treatment with complement inhibitors. Evidence indicates complement activation in patients with HELLP syndrome and in TA-TMA. Early recognition and treatment are essential for patient survival. Complement-targeted therapies show utility in this population, and therefore, further clinical trials and research are needed according to the role of complement in the pathophysiology of SOS/VOD.

## Figures and Tables

**Figure 1 ijms-24-05620-f001:**
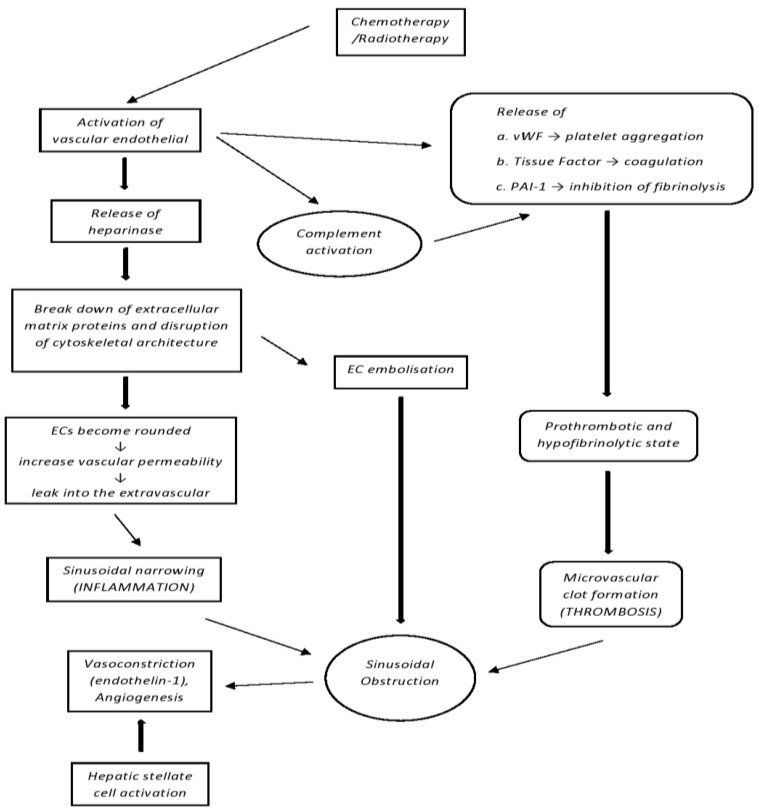
Pathogenesis of SOS/VOD, chemotherapy, and radiation injure endothelial cells (ECs), activate both them and their complement, and disrupt cytoskeletal architecture. Fluid and blood elements enter the extravascular space of Disse and nearby tissues, promoting sinusoidal narrowing, decrease in blood flow, and obstruction.

**Table 1 ijms-24-05620-t001:** Criteria established for diagnosis of SOS patients.

Modified Seattle	Baltimore	EBMT–Classical	EBMT–Late Onset	EBMT–Pediatric
At least 2 of the following before day 20 post-HSCT	Bilirubin ≥ 2 mg/dL before day 21 post-HSCT AND at least 2 of the following:	Bilirubin ≥ 2 mg/dL before day 21 post-HSCT AND at least 2 of the following:	Classical SOS beyond day 21 OR Histologically proven SOS OR At least 2 of the following:	Presence of at least 2 of the following with no limitation for time of onset:
Bilirubin ≥ 2 mg/dL			Bilirubin ≥ 2 mg/dL	Rising bilirubin on at least 3 consecutive days OR Bilirubin ≥ 2 mg/dL within 72 h
Hepatomegaly, RUQ pain	Hepatomegaly	Painful hepatomegaly	Painful hepatomegaly	Hepatomegaly
Ascites with or w/o unexplained weight gain >2% from baseline	Ascites	Ascites	Ascites AND Hemodynamical and/or ultrasound evidence of SOS	Ascites Unexplained consumptive and transfusion-refractory thrombocytopenia
	Weight gain > 5% from baseline	Weight gain > 5% from baseline	Weight gain > 5% from baseline	Unexplained weight gain on 3 consecutive days (despite diuretics) OR Weight gain > 5% from baseline

**Table 2 ijms-24-05620-t002:** New EBMT criteria for severity grading of a suspected SOS/VOD in adult patients.

	Mild	Moderate	Severe	Very Severe-MOD/MOF
**Time since first clinical symptoms of SOS/VOD**	<7 Days	5–7 Days	≤4 Days	Any time
**Bilirubin (mg/dL)**	≥2 and <3	≥3 and <5	≥5 and <8	≥8
**Bilirubin kinetics**			Doubling within 48 h	
**Transaminases**	≤2 × normal	> 2 and ≤5 × normal	>5 and ≤8 × normal	>8 × normal
**Weight increase**	<5%	≥5% and <10%	≥5% and <10%	≥10%
**Renal function**	<1.2 × baseline at transplant	≥1.2 and <1.5 × baseline at transplant	≥1.5 and <2 × baseline at transplant	≥2 × baseline transplant or other signs of MOD/MOF

Multiorgan dysfunction (MOD), multiorgan failure (MOF).

## Data Availability

Not applicable.

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
