# Peer review of "Molecular Advances in Sinusoidal Obstruction Syndrome/Veno-Occlusive Disease"

_ijms, 2023, doi:10.3390/ijms24065620_

Round 1

Reviewer 1 Report

Mavrikou and colleagues provide a short review on molecular advances in SOS/VOD

As a suggestion, I would provide more intel throughout the manuscript on the molecular basis of SOS/VOD pathogenesis, prophylaxis and therapy, in order to comply with the title.

In order to do so, I would expand and deepen every topic touched in the review, expect from the part regarding complement activation that already has a good level of detail.

Other comments:

Abstract: 

-It is difficult to provide novel knowledge with a review, I would change this sentence in the abstract

Introduction:

-HCT abbreviation should be defined

- I suggest discussing the role of microbiome in SOS/VOD pathogenesis (PMID: 34253759). 

- the introduction of defibrotide in the treatment of SOS/VOD has mainly led to the renewed interest in this syndrome, I would rephrase the sentence.

Diagnosis:

-I would describe in more details the severity assessment 

-I would expand the importance of platelets transfusion refractoriness in the diagnosis of SOS/VOD 

Therapy: 

-I would briefly cite the therapeutic choices before defibrotide and expand the description of the potential pharmacological effect

Prophylaxis:

-I would describe the design of the adult randomized trial using defibrotide

Conclusion:

-some sentences need to be re-wrote due to typing errors

The manuscript would benefit from light english editing

Author Response

Reviewer 1

- “Mavrikou and colleagues provide a short review on molecular advances in SOS/VOD

As a suggestion, I would provide more intel throughout the manuscript on the molecular basis of SOS/VOD pathogenesis, prophylaxis and therapy, in order to comply with the title.

In order to do so, I would expand and deepen every topic touched in the review, expect from the part regarding complement activation that already has a good level of detail.

We have provided more details concerning SOS/VOD pathogenesis, prophylaxis and therapy.

Other comments:

Abstract:

-It is difficult to provide novel knowledge with a review, I would change this sentence in the abstract

We have changed this sentence to up-to-date knowledge.

Introduction:

-HCT abbreviation should be defined

HCT is defined.

- I suggest discussing the role of microbiome in SOS/VOD pathogenesis (PMID: 34253759).

It is a very interesting proposal that we have already added to our review.

- the introduction of defibrotide in the treatment of SOS/VOD has mainly led to the renewed interest in this syndrome, I would rephrase the sentence.

 We have rephrased this sentence.

Diagnosis:

-I would describe in more details the severity assessment

We described in more details the severity assessment.

-I would expand the importance of platelets transfusion refractoriness in the diagnosis of SOS/VOD

The importance of platelets transfusion refractoriness in the diagnosis of SOS/VOD is mentioned in the review.

Therapy:

-I would briefly cite the therapeutic choices before defibrotide and expand the description of the potential pharmacological effect

We mentioned therapeutic choices before defibrotide.

Prophylaxis:

-I would describe the design of the adult randomized trial using defibrotide

We have described an adult randomized trial using defibrotide.

Conclusion:

-some sentences need to be re-wrote due to typing errors

We examined the paper and re-wrote the sentences.

The manuscript would benefit from light english editing.”

We understand the reviewer’s concern.

Thank you very much for your review.

Reviewer 2 Report

The presented article consists literature review of SOS/VOD in liver disorders. 

The article needs to be revised and needs to be improved. Please rearrange the article for better understanding and make it easier to follow the researcher's thoughts. Also, the title needs to be commensurate with the content. The authors need to indicate which "molecular advances" are known and which need to be better understood. 

 Literature concerning the analyzed factors is nicely cited.

Moreover, there needs to be added some information how publications were selected in the databases (what criteria were taken into account). 

Details:

Line 50-52

"The mortality rate of severe SOS/VOD is high (...)" - please add more details here (for one, the high is ten while for the others, it is hundreds). 

Line 63- 65

The Authors could add here some information about how many papers have been published at this time. 

Author Response

Reviewer 2

“The presented article consists literature review of SOS/VOD in liver disorders.

The article needs to be revised and needs to be improved. Please rearrange the article for better understanding and make it easier to follow the researcher's thoughts. Also, the title needs to be commensurate with the content. The authors need to indicate which "molecular advances" are known and which need to be better understood.

We have indicated which molecular advances are known and which need to be better understood.

 Literature concerning the analyzed factors is nicely cited.

Moreover, there needs to be added some information how publications were selected in the databases (what criteria were taken into account).

We have added all the information that concern the publications and their selection.

Details:

Line 50-52

"The mortality rate of severe SOS/VOD is high (...)" - please add more details here (for one, the high is ten while for the others, it is hundreds).

We have added more details.

Line 63- 65

The Authors could add here some information about how many papers have been published at this time.”

We have added information about the number of papers that have been published.

Thank you very much for your review.

Reviewer 3 Report

This manuscript reviews sinusoidal obstruction syndrome/veno-occlusive disease (SOS/VOD) detected in the liver, which is a severe complication of hematopoietic stem cell transplantation (HSCT). Antibodies conjugated with calicheamicin, gemtuzumab and inotuzumab ozogamicin have renewed interest in SOS/VOD; however, characterizing high-risk patients for SOS/VOD is still needed. Here, the authors summarize current molecular advances on diagnosis and treatment of SOS/VOD. Overall, this review provides up-to-date knowledge on genetic and other markers for SOS/VOD for patient risk stratification and should be of some interest to members of the field.

Author Response

Reviewer 3

“This manuscript reviews sinusoidal obstruction syndrome/veno-occlusive disease (SOS/VOD) detected in the liver, which is a severe complication of hematopoietic stem cell transplantation (HSCT). Antibodies conjugated with calicheamicin, gemtuzumab and inotuzumab ozogamicin have renewed interest in SOS/VOD; however, characterizing high-risk patients for SOS/VOD is still needed. Here, the authors summarize current molecular advances on diagnosis and treatment of SOS/VOD. Overall, this review provides up-to-date knowledge on genetic and other markers for SOS/VOD for patient risk stratification and should be of some interest to members of the field.”

Thank you very much for your review.

Round 2

Reviewer 1 Report

The authors addressed some of my concerns

However, there are still few insights throughout the test regarding molecular advances on SOS/VOD and the authors could be more precise in the description of new EBMT criteria, platelet refractoriness and on the role of Microbiome. 

Moreover, English editing is still needed before publication and I truly suggest the author to perform English revision from a professional.

Author Response

Reviewer 1

The authors addressed some of my concerns

However, there are still few insights throughout the test regarding molecular advances on SOS/VOD and the authors could be more precise in the description of new EBMT criteria, platelet refractoriness and on the role of Microbiome. 

  • We added more insights throughout the test regarding molecular advances on SOS/VOD. Furthermore, we are more precise concerning the description of new EBMT criteria and we demonstrate the recently published severity scoring system of SOS/VOD in adult patients in Table 2. We mention platelet refractoriness as an early diagnostic criterion in the 2017 EBMT criteria in Table 1. Moreover, we cite the association of gut microbiome with HSCT outcomes.

Additionally, English editing is still needed before publication and I truly suggest the author to perform English revision from a professional.

  • The manuscript has been reviewed by a native English speaker, Margarita Sakellari.

Extensive editing of English language and style required

  • The manuscript has been reviewed by a native English speaker, Margarita Sakellari. 

Reviewer 2 Report

The authors did not thoroughly revise the article. Still, there is much more work to make this paper more to make the article more readable for the readers. 

Author Response

Reviewer 2

The authors did not thoroughly revise the article. Still, there is much more work to make this paper more to make the article more readable for the readers. 

  • We have thoroughly revised the article in order to make the paper more readable for the readers.

 English language and style are fine/minor spell check required

  • The manuscript has been reviewed by a native English speaker, Margarita Sakellari.

Round 3

Reviewer 2 Report

This manuscript presents a substantial improvement.